# Widespread Detection of Yersiniabactin Gene Cluster and Its Encoding Integrative Conjugative Elements (ICEKp) among Nonoutbreak OXA-48-Producing *Klebsiella pneumoniae* Clinical Isolates from Spain and the Netherlands

Afif P. Jati,[a,b] Pedro J. Sola-Campoy,[c] Thijs Bosch,[d] Leo M. Schouls,[d] Antoni P. A. Hendrickx,[d] Verónica Bautista,[c] Noelia Lara,[c] Erwin Raangs,[a] Belén Aracil,[c,e] John W. A. Rossen,[a,f,g] Alex W. Friedrich,[a,h] Ana M. Navarro Riaza,[c] Javier E. Cañada-García,[c] Eva Ramírez de Arellano,[c,e] ![ORCID] Jesús Oteo-Iglesias,[c,e] María Pérez-Vázquez,[c,e] ![ORCID] Silvia García-Cobos[a,c]
**The Dutch and Spanish Collaborative Working Groups on Surveillance on Carbapenemase-Producing Enterobacterales**

[a]University of Groningen, University Medical Center Groningen, Department of Medical Microbiology and Infection Prevention, Groningen, The Netherlands
[b]Indonesian Society of Bioinformatics and Biodiversity, Indonesia
[c]Laboratorio de Referencia e Investigación en Resistencia a Antibióticos e Infecciones Relacionadas con la Asistencia Sanitaria, Centro Nacional de Microbiología, Instituto de Salud Carlos III, Majadahonda, Madrid, Spain
[d]Infectious Diseases Research, Diagnostics and Laboratory Surveillance, Centre for Infectious Disease Control Netherlands, National Institute for Public Health and the Environment, Bilthoven, The Netherlands
[e]CIBER de Enfermedades Infecciosas, Spanish Network for Research in Infectious Diseases, Instituto de Salud Carlos III, Madrid, Spain
[f]Laboratory of Medical Microbiology and Infectious Diseases, Isala Hospital, Zwolle, The Netherlands
[g]Department of Pathology, University of Utah School of Medicine, Salt Lake City, Utah, USA
[h]University Hospital Münster, Institute of European Prevention Networks in Infection Control, Münster, Germany

**ABSTRACT** In this study, we determined the presence of virulence factors in non-outbreak, high-risk clones and other isolates belonging to less common sequence types associated with the spread of OXA-48-producing *Klebsiella pneumoniae* clinical isolates from The Netherlands ($n = 61$) and Spain ($n = 53$). Most isolates shared a chromosomally encoded core of virulence factors, including the enterobactin gene cluster, fimbrial *fim* and *mrk* gene clusters, and urea metabolism genes (*ureAD*). We observed a high diversity of K-Locus and K/O loci combinations, KL17 and KL24 (both 16%), and the O1/O2v1 locus (51%) being the most prevalent in our study. The most prevalent accessory virulence factor was the yersiniabactin gene cluster (66.7%). We found seven yersiniabactin lineages—*ybt* 9, *ybt* 10, *ybt* 13, *ybt* 14, *ybt* 16, *ybt* 17, and *ybt* 27—which were chromosomally embedded in seven integrative conjugative elements (ICEKp): ICEKp3, ICEKp4, ICEKp2, ICEKp5, ICEKp12, ICEKp10, and ICEKp22, respectively. Multidrug-resistant lineages—ST11, ST101, and ST405—were associated with *ybt* 10/ICEKp4, *ybt* 9/ICEKp3, and *ybt* 27/ICEKp22, respectively. The fimbrial adhesin *kpi* operon (*kpiABCDEFG*) was predominant among ST14, ST15, and ST405 isolates, as well as the ferric uptake system *kfuABC*, which was also predominant among ST101 isolates. No convergence of hypervirulence and resistance was observed in this collection of OXA-48-producing *K. pneumoniae* clinical isolates. Nevertheless, two isolates, ST133 and ST792, were positive for the genotoxin colibactin gene cluster (ICEKp10). In this study, the integrative conjugative element, ICEKp, was the major vehicle for yersiniabactin and colibactin gene clusters spreading.

**IMPORTANCE** Convergence of multidrug resistance and hypervirulence in *Klebsiella pneumoniae* isolates has been reported mostly related to sporadic cases or small outbreaks. Nevertheless, little is known about the real prevalence of carbapenem-resistant hypervirulent *K. pneumoniae* since these two phenomena are often separately studied. In this study, we gathered information on the virulent content of nonoutbreak, high-risk clones (i.e., ST11, ST15, and ST405) and other less common STs

**Ad Hoc Peer Reviewer** ![ORCID] Andrés Marcoleta, Universidad de Chile

Address correspondence to Silvia García-Cobos, s.garciacobos@isciii.es.

The authors declare a conflict of interest. J.W.A.R. was employed by IDbyDNA and is currently consulting for ARES-genetics. The remaining authors declare that the research was conducted in the absence of any commercial or financial relationships that could be construed as a potential conflict of interest.

associated with the spread of OXA-48-producing *K. pneumoniae* clinical isolates. The study of virulence content in nonoutbreak isolates can help us to expand information on the genomic landscape of virulence factors in *K. pneumoniae* population by identifying virulence markers and their mechanisms of spread. Surveillance should focus not only on antimicrobial resistance but also on virulence characteristics to avoid the spread of multidrug and (hyper)virulent *K. pneumoniae* that may cause untreatable and more severe infections.

**KEYWORDS** *Klebsiella pneumoniae*, hypervirulence, virulence, carbapenem resistance, $bla_{OXA-48}$, OXA-48, antibiotic resistance, carbapenemase, virulence factors, whole-genome sequencing

**K**lebsiella pneumoniae is a significant cause of severe hospital and community-acquired infections. The surveillance of the emergence of carbapenem-resistant *K. pneumoniae* is a priority for public health organizations. The class D carbapenemase OXA-48 is an efficient enzyme to hydrolyze imipenem and frequently shows a "mask" phenotype that makes it difficult to diagnose, i.e., OXA-48-producing strains can exhibit low-level carbapenem resistance, as can extended-spectrum $\beta$-lactamase (ESBL)-producing strains with decreased permeability (1). Moreover, the OXA-48 enzyme is frequently coproduced with other antibiotic resistance genes either carbapenemases or ESBLs (2, 3).

A highly transferable IncL group plasmid (pOXA-48a) is responsible for the spread of the $bla_{OXA-48}$ gene in *K. pneumoniae* (4, 5), which is often associated with successful lineages such as ST11 and ST405. OXA-48-producing *K. pneumoniae* (OXA-48-*Kp*) has been implicated in hospital outbreaks and even recently among COVID-infected patients (6) and has been documented in many countries (7), being one of the most prevalent carbapenemases in Europe (8).

The widespread dissemination of some lineages and high-risk clones raises the question of what genetic traits could enhance their infectiveness. *K. pneumoniae* presents a variety of sophisticated immune evasion strategies and virulence factors. However, they have not been systematically investigated in high-risk clones, bed-side-patient treatment primarily depends on antibiotic resistance phenotypes, and the characterization of the vast virulent content can be laborious using traditional techniques. The investigation of *K. pneumoniae* molecular pathogenesis becomes even more challenging due to the horizontal gene transfer of mobile genetic elements (MGEs) such as virulence plasmids, integrative and conjugative elements (ICEs), and genomic islands (9–12). ICEs are chromosomally located gene clusters that encode phage-linked integrases, conjugation proteins, and other genes associated with virulence or resistance and which can be transferred between cells. On the contrary, genomic islands have not been shown to transfer (9). Whole-genome sequencing (WGS) data allows rapid screening of the bacterial genomic pathogenicity and particularly third-generation (long-read) sequencing allows a deeper analysis of the MGEs involved.

The definition of hypervirulent *K. pneumoniae* (hv*KP*) is still not fully agreed upon and cannot be attributed to a single factor due to the diversity and complexity of possible virulence mechanisms. Up to now, some well-characterized phenotypes and factors serve as markers for hv*KP*, i.e., hypermucoviscosity, K1/K2 locus types, aerobactin siderophore, and virulence plasmids (13). In addition to the phenotype and genetic background, clinical manifestations can be considered, such as major invasiveness and additional complications (14, 15).

Traditionally, multidrug resistance and hypervirulence have been two nonoverlapping phenotypes for *K. pneumoniae*, associated with distinct clonal lineages (16). However, carbapenem-resistant hypervirulent *K. pneumoniae* (CR-hv*KP*) prevalence increased in the last decade due to three different phenomena. The reasons for this, from the more likely to the less likely, are as follows: (i) classical carbapenem-resistant *K. pneumoniae* lineages (i.e., ST11/ST258) acquire a virulence plasmid; (ii) classical hypervirulent lineages (i.e., K1/K2/K5

locus types or ST23/ST86/ST65) acquire plasmids harboring carbapenemase genes or even develop point mutations on the chromosome; and (iii) *K. pneumoniae* acquires a hybrid plasmid combining both carbapenem resistance and hypervirulence (17).

In this study, we aimed to assess the genomic pathogenicity profiles of OXA-48-producing *K. pneumoniae* clinical isolates collected from Spain and the Netherlands. We studied the capsule polysaccharide, lipopolysaccharide, fimbriae, siderophores, and other virulence factors. We also investigated the genetic location of the studied virulence factors, antimicrobial resistance genes, and plasmid replicon genes to raise awareness of the concurrence of antibiotic resistance and virulence in OXA-48-producing *K. pneumoniae* clinical isolates.

## RESULTS AND DISCUSSION

We characterized the genetic virulence profile of nonoutbreak OXA-48-producing *K. pneumoniae* clinical isolates from Spain and the Netherlands based on the presence of core and accessory virulence factors and revealed their genetic location, plasmid or chromosomal, to assess their spread mechanisms and their concurrence with antibiotic-resistant traits. After contamination analysis of short reads using Mash Screen, all isolates confirmed as *K. pneumoniae* subsp. *pneumoniae* were selected, including 53 isolates from Spain and 61 from the Netherlands (see Table S1 at https://doi.org/10.6084/m9.figshare.22794398.v2).

**Genetic relatedness of OXA-48-producing *K. pneumoniae* clinical isolates from the Netherlands and Spain. (i) Close genetic relatedness (complex types, cgMLST threshold ≤15 allele differences).** The OXA-48- producing *K. pneumoniae* clinical isolates were selected considering nonoutbreak relationship. Nevertheless, we analyzed the genetic relatedness of our *K. pneumoniae* collection using core genome multilocus sequence typing (cgMLST) and core genome single nucleotide polymorphism (cgSNP) approaches to confirm enough genetic variability and to investigate virulent patterns within *K. pneumoniae* population. First, using a ≤15 allele difference threshold in the cgMLST analysis, we observed 12 groups or complex types (CTs) of isolates and 74 singletons. No CT was composed of isolates from both countries. All CTs were formed by isolates from different hospitals, except one (CT10) in which two isolates were from the same Dutch hospital but isolated in different years (2016 and 2017) (Fig. 1A; see also Table S3 at https://doi.org/10.6084/m9.figshare.22794398.v2). Second, we investigated the number of cgSNPs between isolates in each complex type obtained by the cgMLST analysis (≤15 allele difference threshold) (Fig. 2; see also Table S3 and Data Set S3 at https://doi.org/10.6084/m9.figshare.22794233.v2). In all CTs, isolates had ≥8 cgSNPs, except four pair genomes with two to five cgSNPs: ISC13 and ISC39 (ST11, two cgSNPs, CT1), 544789 and 544803 (ST16, three cgSNPs, CT9), ISC19 and ISC55 (ST15, four cgSNPs, CT6), and ISC19 and ISC45 (ST15, five cgSNPs, CT6). It is noteworthy that some of these STs—ST11 and ST15—are recognized as global multidrug-resistant (MDR) clones (18). Furthermore, ST11 has been circulating in Spain since the first reported case of OXA-48-producing *K. pneumoniae* in 2009 (19, 20) and is found worldwide (4).

**(ii) Genomic population structure (clonal groups, cgMLST threshold ≤43 allele differences, and sublineages, cgMLST threshold ≤190 allele differences).** Considering *K. pneumoniae* population structure and following Hennart et al. (21) approach, we analyzed the distribution of the number of allelic differences among all pairs of genomes. We observed 17 CGs, several of the same STs, such as CG101 and CG15, and larger SL groups (Fig. 1B and C; see also Data Set S4 at https://doi.org/10.6084/m9.figshare.22794269.v2).

**K and O loci of OXA-48-*Kp* from the Netherlands and Spain.** The K-Locus capsular polysaccharide (CPS) and the O-Locus lipopolysaccharide (LPS) are important determinants of virulence and bacterial interaction with the immune system (22). During the infection, the capsule helps *K. pneumoniae* to escape phagocytosis from neutrophils and macrophages (12). We identified 29 different K loci and 9 O loci, and these K/O loci provided 31 different combinations in our OXA-48-*Kp* collection (Fig. 2; see also Fig. S2 at https://doi.org/10.6084/m9.figshare.22794050.v3 and Fig. S3 at https://doi.org/10.6084/m9.figshare.22794053.v3). KL17 (16%), KL24 (16%), and KL151 (15%) were the

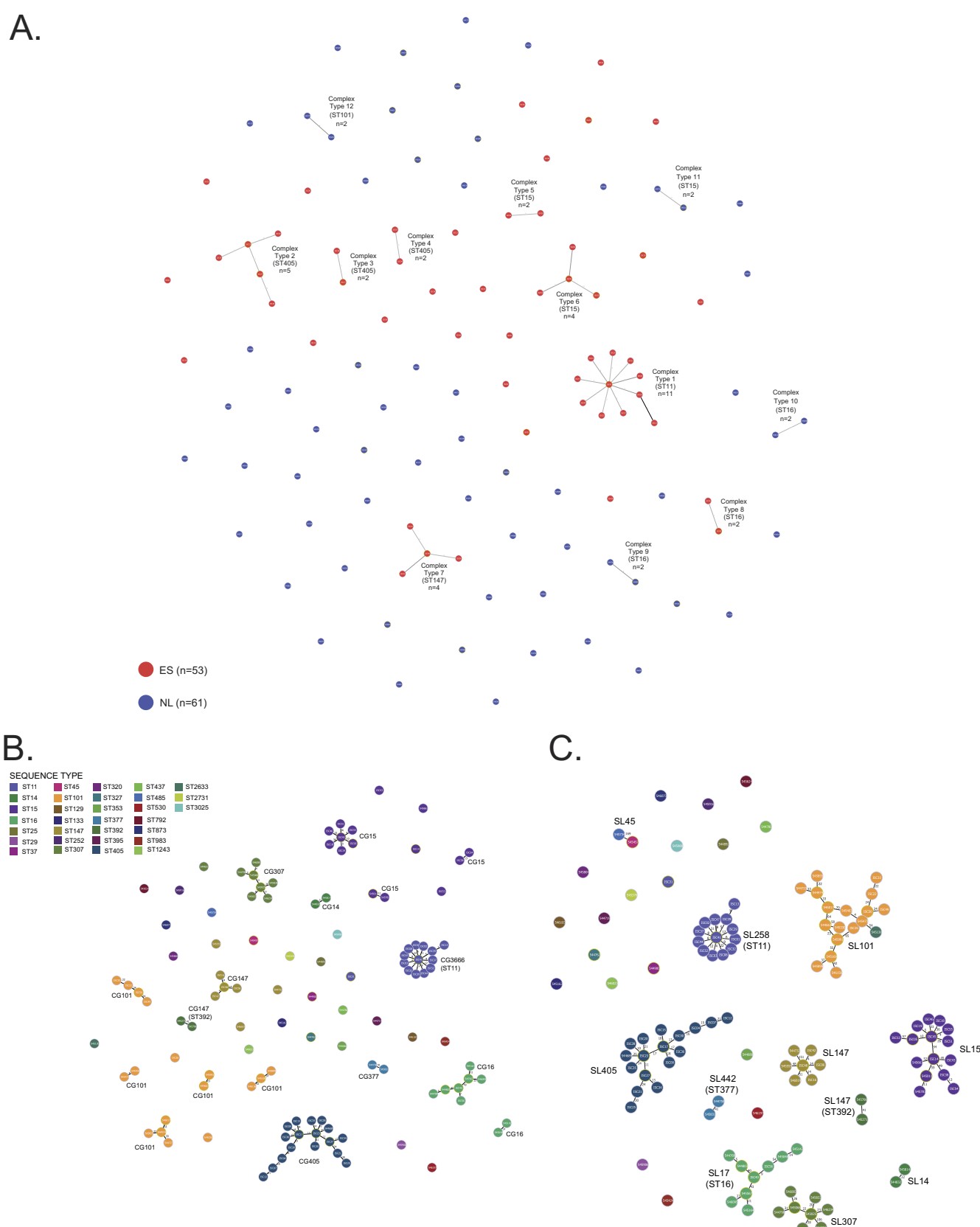

**FIG 1** Minimum spanning tree of 114 OXA-48-producing *K. pneumoniae* clinical isolates from Spain and the Netherlands created using PHYLOViZ 2 (72). Each circle represents an allelic profile based on a cgMLST scheme of 2,365 target genes (SeqSphere+ Ridom, GmbH, Münster, Germany). (A) Complex types based on a threshold of ≤15 allele differences. Colors indicate the country of origin (red, Spain; blue, the Netherlands). (B) Clonal groups (CGs) based on a threshold of ≤43 allelic differences, colored by STs. (C) Sublineages (SLs) based on a threshold of 190 allelic differences, colored by STs.

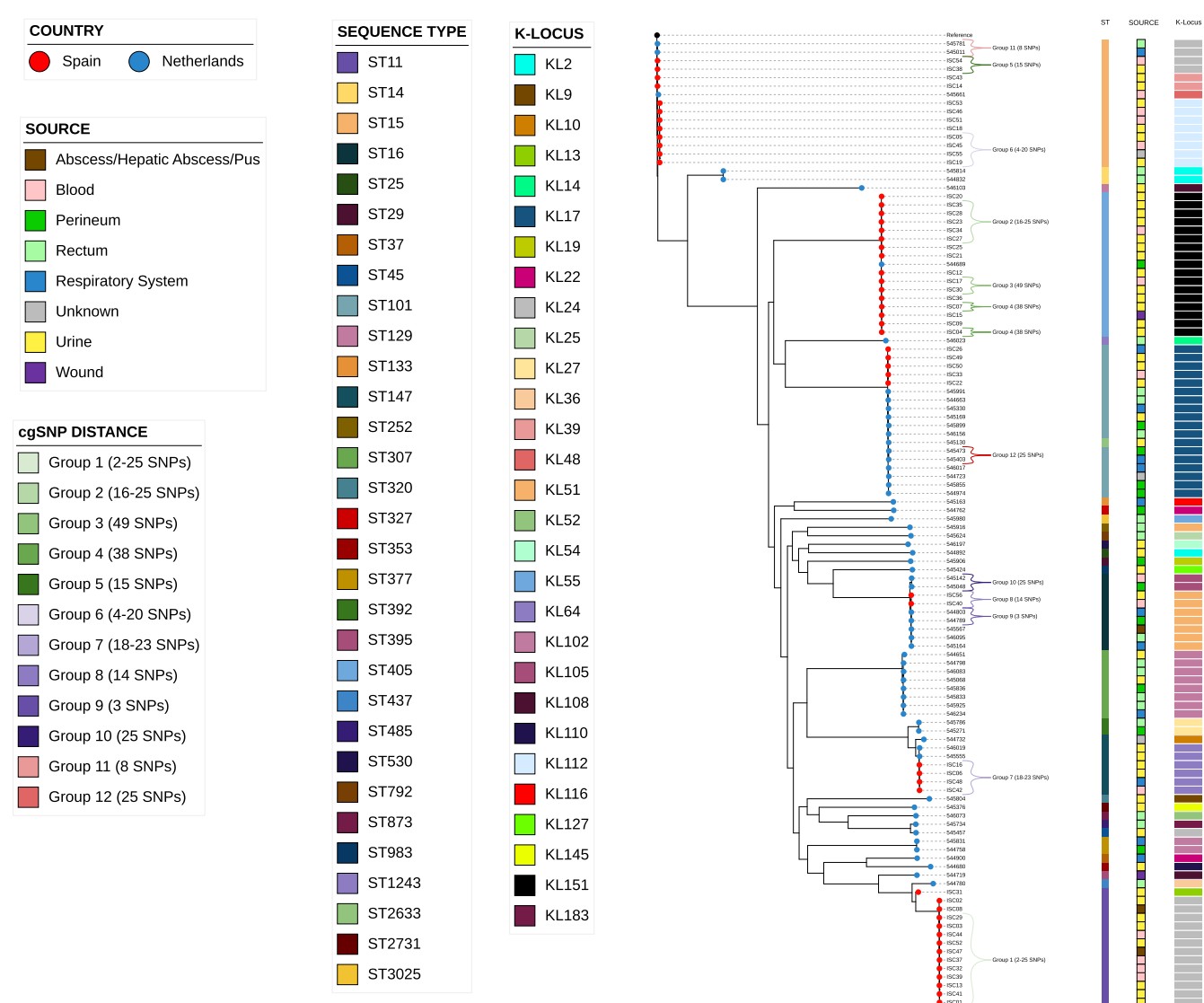

**FIG 2** Diversity of K loci among 114 OXA-48-producing *K. pneumoniae* clinical isolates and associated STs. A maximum-likelihood tree based on core-genome SNP analysis using an alignment of 126,803 positions is shown. Groups are referred to the twelve Complex Types (CTs) formed when using a threshold of ≤15 allele diferences in the cgMLST analysis.

most prevalent K loci. The higher diversity of the K-Locus in *K. pneumoniae* compared to the O-Locus is due to a higher level of diversity in both sequence and gene content in the *cps* cluster (23, 24). Among the predominant STs we observed the following K+O loci: KL17+O1/O2v1 (16%, 18/114) associated with ST101 (*n* = 17) and ST2633 (*n* = 1); KL24+O1/O2v1 (16%, 18/114) associated with ST11 (*n* = 13), ST15 (*n* = 4), and ST45 (*n* = 1); KL151+O4 (15%, 17/114) associated with ST405; and KL102+O1/O2v2 (9%, 10/114) associated with ST307 (*n* = 8) and ST377 (*n* = 2) (see Fig. S4 at https://doi .org/10.6084/m9.figshare.22794047.v3).

Horizontal gene transfer of the *cps* operon or evolutionary convergence is likely to affect K-Locus diversity, causing identical K loci among unrelated genomic background isolates (25). We observed this phenomenon in our collection of isolates, e.g., KL24 was observed in ST11, ST15, and ST45 isolates. In addition, this high K/O loci diversity seems to be more associated with MDR clones than with hypervirulent ones due to a significant recombination tendency at the capsule (K) and adjacent LPS antigen (O) in MDR clones (26). On the other hand, we observed the same K-Locus combined with a

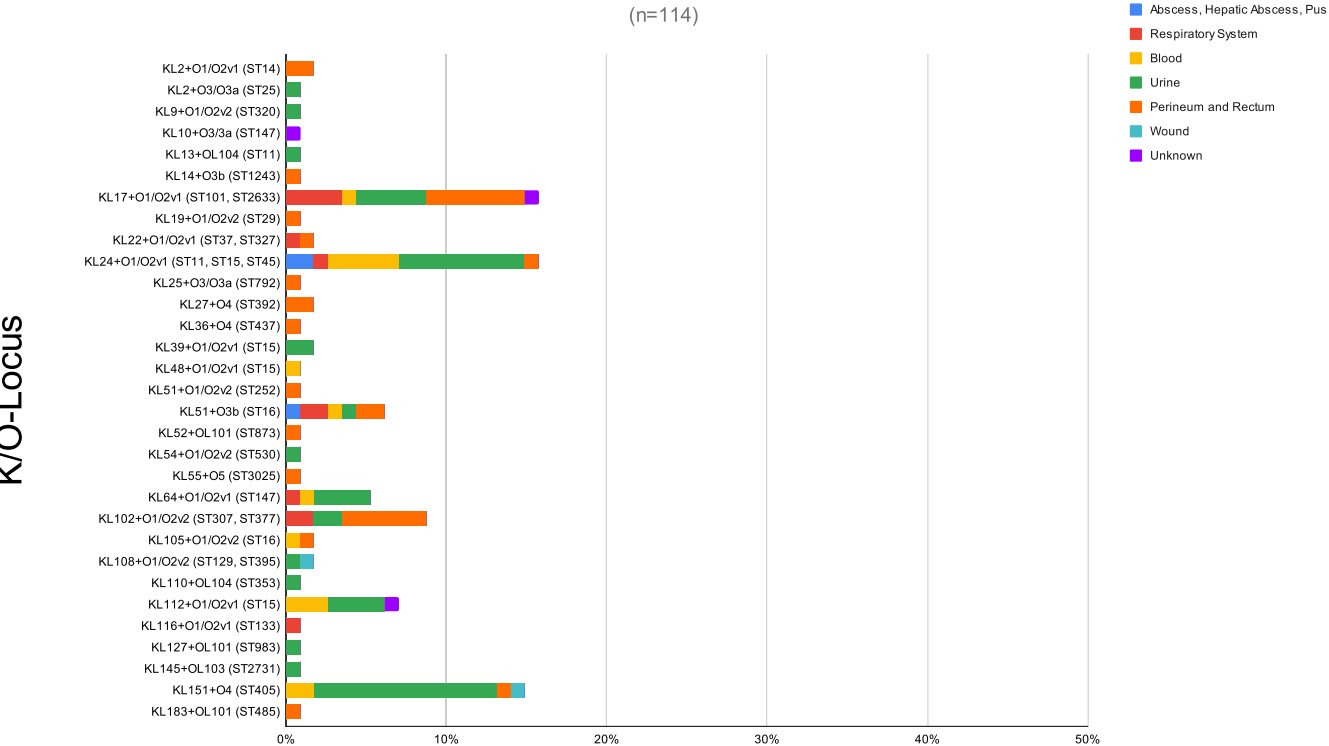

**FIG 3** Distribution of K and O loci according to the source of isolation of 114 OXA-48-producing *K. pneumoniae* clinical isolates.

distinct O-Locus, i.e., KL2 associated with O1/O2v1 (ST14) and O3/3a (ST25) and KL51 linked to O1/O2v2 (ST252) and O3b O-loci (ST16), as well as the same O-Locus combined with a distinct K-Locus. This phenomenon has been related to the reassortment of K and O loci in *K. pneumoniae* (23).

This high diversity of the K-Locus and combinations of K and O loci hampers the prediction of clone identity based on the K-Locus (25, 27) and makes it challenging to establish an association with infection site or type of disease as previously described (23). Nevertheless, specific K loci, KL1 and KL2, have been associated with unfavorable disease outcomes and invasive infections (i.e., pyogenic liver abscess in Asia) (18, 23). In our collection, three KL2 isolates (one ST25 and two ST14) were not associated with invasive infections but with perineum/rectum and urine origins (Fig. 2 and 3). These isolation sites should not be undervalued since intestinal colonization has been described as an initial step for infection (14, 28). We also found one isolate with the KL54+O1/O2v2 loci (ST530) originating from urine. KL54 has previously been described as a hypervirulent K antigen because of its fucose-based capsular type, which is unrecognizable to the immune system and causes severe infections (29).

Whereas a high diversity of K loci exists, only a few O loci (*rfb* locus) have been reported, with O1, O2, and O3 loci being dominant in human disease (23). Although the majority of LPS O antigens are associated with unique O loci, some exceptions associated with either two distinct O loci have been described (30). This is the case in our study, being O1/O2v1 loci (51%) the most prevalent, from perineum, rectum, urine, and blood origin isolates (see Fig. S5 at https://doi.org/10.6084/m9.figshare.22794044.v3), which has been described contributing to invasive tissue infection (pyogenic liver abscess) and playing an important role in bacterial dissemination and colonization of internal organs (31). Within our collection, one isolate (ST11, KL24, and O1/O2v1) originated from hepatic abscess, but it did not have any of the previously described genetic determinants for hypervirulence: *rmpA*, aerobactin, *kfu*, *allS*, and KL1/KL2 K loci (32). However, we did not have information on the hypermucoviscosity phenotype of this isolate, which has been

described as a virulence determinant of the liver abscess causative isolates regardless of any capsular K loci (32).

The second most common O-Locus in our collection, the O4 locus (18%), was predominantly found in urine samples and mostly belonged to ST405 (see Fig. S3), followed by the O1/O2v2 locus (16%), which was distributed within isolates from perineum, rectum and urine samples (see Fig. S5). This O-Locus has been broadly associated with MDR clones and ESBL-producing and carbapenem-resistant bacteria (33). In addition, the O3/3a locus, rarely found in clinical isolates (34), was observed in one isolate belonging to ST792, KL25, that had the colibactin gene cluster (see Fig. S3).

**Core virulence factors in OXA-48-producing *K. pneumoniae* clinical isolates.** In addition to the capsular polysaccharide (K antigen) and LPS (O antigen) biosynthesis loci, a subset of core chromosomally encoded virulence factors has been described as required for establishing opportunistic infections in mammalian hosts: the siderophore enterobactin and type 1 and type 3 fimbriae (18).

Enterobactin is a highly conserved siderophore in *Klebsiella* spp. population, both in classical and hypervirulent strains, so that neutrophils and mucosal surfaces produce the innate immune protein Lipocalin 2 (Lcn2) able to bind it, preventing bacteria from iron acquisition (35, 36). We detected the enterobactin gene cluster (*fepABCDG* and *entBCDEF*) in all isolates (Fig. 4) (gene *entD* had a 75% coverage).

Highly conserved type 1 (*fim*) and type 3 (*mrk*) fimbriae gene clusters are important for adhesion to biotic and abiotic surfaces (12). We identified the type 1 fimbrial gene cluster (*fimABCDEFGHIK*) complete in 91.2% (104/114) of the isolates; eight isolates had type 1 fimbriae gene cluster incomplete, and it was absent in two isolates. Regarding the type 3 fimbrial gene cluster (*mrkABCDFHIJ*), it was complete in 90.4% (103/114) of the isolates; nine isolates had type 3 fimbriae gene cluster incomplete, and it was absent in two isolates (Fig. 4).

Type 1 fimbriae are a significant virulence factor in *K. pneumoniae* urinary tract infections (37), and both type 1 and type 3 fimbriae are expressed during biofilm formation on urinary tract catheters (38, 39). In our collection, these two fimbrial gene clusters were present in isolates from diverse origins, such as urine, blood, and sputum. The spread of type 3 fimbrial genes by lateral gene transfer has been documented (40).

In addition, the *yfiN* gene, involved in the expression of type 1 and type 3 fimbriae (38, 41, 42), was also present in all isolates in this collection and is encoded in an operon that regulates extracellular polymeric substance production in *K. pneumoniae* (43).

Furthermore, and although not previously defined as core virulence genome, we detected *wabG* and *uge* genes, associated with core LPS synthesis, in 100% (114/114) and 99% (113/114) of the isolates, respectively. *ureA* and *ureD* genes, which encode urease activity and are important for urea metabolism, as well as the adhesin gene *ycfM*, were also present in all isolates (Fig. 4). During the early stage of nosocomial infection, *K. pneumoniae* colonizes the gastrointestinal tract, and urease helps the cell resist gastrointestinal stress (44). Notably, gastrointestinal colonization is a primary source of *K. pneumoniae* infection in Intensive Care Unit patients (45). The ubiquity of the *ycfM* gene in the *K. pneumoniae* genome has been previously reported (46). This gene participates in the production of outer membrane lipoprotein and is recognized as a nonfimbrial adhesin and a putative fibronectin-binding protein that facilitates adhesion to abiotic surfaces (46). Thus, the high prevalence of genes *wabG*, *uge*, *ureA*, *ureD*, and *ycfM* highlights their possible role as core virulence genomes in *K. pneumoniae*.

**Accessory virulence factors in OXA-48-producing *K. pneumoniae* clinical isolates.** Virulence genes with a variable presence are referred to as accessory genomes. This includes chromosomally integrated and plasmid-based genes, which are transferred through pathogenicity islands (PAIs), and other MGEs (36).

**(i) Chromosomally integrated.** Virulence genes related to the ferric uptake system, *kfuABC*, were present in 48% (55/114) of the isolates, including some MDR STs such as ST14, ST15, ST101, and ST405 (Fig. 4). This gene cluster has been previously described as conserved in hypervirulent clonal group 23 (CG23) (comprising ST23, ST26, ST57, and ST163) (47) but not in other hypervirulent groups (48).

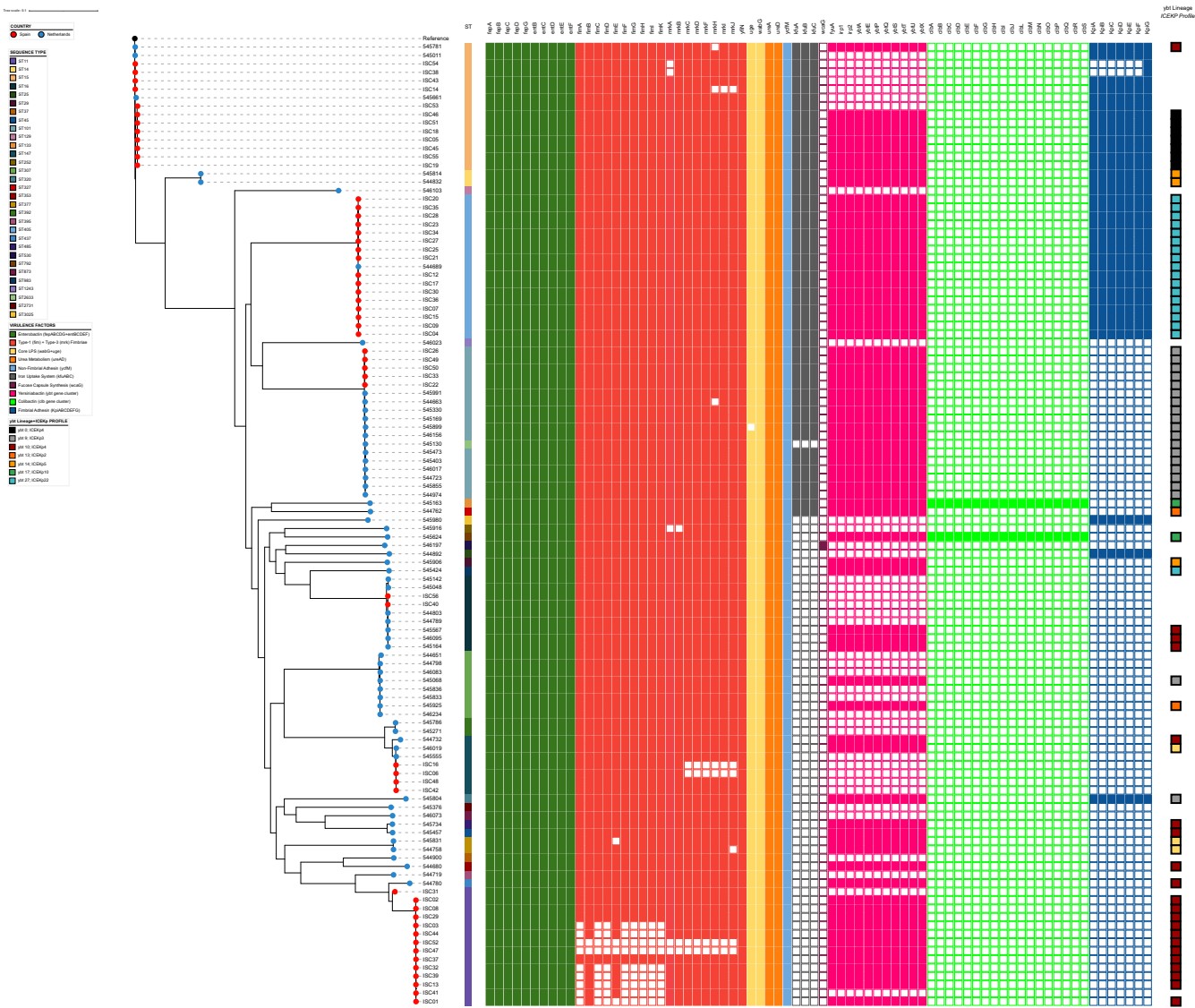

**FIG 4** Dendrogram of maximum-likelihood tree based on cgSNP analysis and a presence/absence heatmap of virulence factors (in-house database) of 114 OXA-48-producing *K. pneumoniae* clinical isolates. Yersiniabactin (*ybt*) lineage and ICE*Kp* combination profiles (Kleborate results) are summarized in the right column.

The *kpi* operon (*kpiABCDEFG*) encodes additional fimbrial adhesin proteins found in 33% (38/114) of the total collection. This operon has been reported to promote host colonization and persistence in the hospital environment (49). A total of 36 isolates carried the complete set of the operon gene cluster except for two ST15 isolates, ISC38 and ISC54, which carried only the *kpiG* gene encoding fimbrial usher protein. Importantly, the *kpi* operon has been described as associated with the worldwide-disseminated ST15 clone (49). Our results confirmed this, being *kpi* operon present in all ST14 and ST15 isolates (except for two isolates that only had the *kpiG* gene) and also in all ST405 isolates (Fig. 4).

The *allS* gene, associated with allantoin metabolism, was absent in all isolates, and the *wcaG* gene, which enables the isolate to create a fucose-based capsule, was only present in one isolate from the Netherlands associated with the hypervirulent KL54 locus type (ST530, urine origin), as previously described, in addition to other K-Locus types (18, 29). The presence of *wcaG* as a component in *cps* could enhance the ability to evade phagocytosis (50), overproduce biofilm, and cause severe infection, such as bacteremia (51).

**(ii) MGE mediated.** Siderophores such as plasmid-mediated salmochelin and aerobactin, hypermucoviscous-associated genes *rmpADC*, *rmpA2* and *magA*, are relevant in virulence, as are the PAI-derived yersiniabactin and genotoxin colibactin gene clusters.

Yersiniabactin and salmochelin are important as alternative strategies for iron acquisition; likewise, with salmochelin, when enterobactin has been inhibited by the host protein Lipocalin 2 (Lcn2) (35, 52), 66.7% (76/114) of the isolates were positive for the yersiniabactin gene cluster. The presence of yersiniabactin has been linked with promoting respiratory tract infections (35). In our collection, the yersiniabactin gene cluster was present in isolates of respiratory origin but also in isolates from urine (43.4%, 33/76), rectum/perineum (23.7%, 18/76), and blood origin (13.2%, 10/76).

Further analysis using the Kleborate tool revealed the genetic diversity of yersiniabactin-encoding integrative conjugative element (ICE*Kp*) (17, 47) and their lineages in our *K. pneumoniae* collection (Fig. 4; see also Data Set S5 at https://doi.org/10.6084/m9.figshare.22794290.v2). The majority of yersiniabactin positive (*ybt*$^+$) isolates, 36.8% (28/76), were spread via ICE*Kp4* related to *ybt* 10 lineage or unassigned *ybt* lineage (*ybt* 0), corresponding mostly to isolates from ST11 (Spain [ES], 15.8%, $n = 12/76$) and ST15 (ES, 9.21%, $n = 7/76$). ICE*Kp4* has been described as a common type among *K. pneumoniae* genomes (53). Second, 26.3% (20/76) *ybt*$^+$ isolates revealed an ICE*Kp3* (*ybt* 9 lineage), associated mostly with isolates from ST101 (ES, 6.6%, $n = 5/76$; Netherlands [NL], 15.8%, $n = 12/76$). ICE*Kp3*, the second most common ICE*Kp* in our collection, could enhance the virulence of *K. pneumoniae* clinical strains (29, 54) and is also associated with hypervirulent strains isolated from surgical sites (55). We also identified 23.7% (18/76) of ICE*Kp22* (*ybt27* lineage) as the third major ICE*Kp* in our collection, which was mostly associated with ST405 isolates. Other MGEs and *ybt* lineages were detected with low frequency (<10%): *ybt* 13/ICE*Kp2*, *ybt* 14/ICE*Kp5*, and *ybt* 16/ICE*Kp12* (Fig. 4).

Two yersiniabactin-positive isolates also harbored the genotoxin colibactin, another PAI-derived gene cluster, associated with ICE*Kp10* (Fig. 4), which is considered a marker of hypervirulence for carrying the *clb* locus (18, 53). The two ICE*Kp10* had the same genetic structure but were inverted. This structure has been previously described (53), with a Zn$^{+2}$/Mn$^{+2}$ metabolism module next to the yersiniabactin gene cluster, and this one separated to the colibactin gene cluster by a mobilization module (Fig. 5).

The colibactin-positive isolates revealed the same *clb3* lineage, but two different CbSTs, 19-1LV and 15-1LV, and were associated with KL25 (ST792) and KL116 (ST133), respectively. ST133/K116:O1/O2v1 *K. pneumoniae* isolates positive for colibactin from sputum origin have been described in Australia and the United States, but these isolates did not have any carbapenemase (26). A *K. pneumoniae* strain producing KPC-2, ST792/KL25, positive for *ybt* and *clb* was described in a study in Singapore (56). The colibactin-positive isolates from our collection were isolated from the perineum/rectum and the respiratory system. This toxin is synthesized by polyketide synthases (*pks*) encoded by a genomic island whose acquisition is associated with *K. pneumoniae* gut colonization and mucosal invasion (57, 58). Furthermore, this toxin could be a potential biomarker for developing life-threatening diseases, such as colorectal cancer (58) and meningitis (59). These associations are based on the high prevalence of *pks*-positive *E. coli* and *pks*-positive *K. pneumoniae* observed in colorectal patient samples, its role in the induction of mutations in colorectal cancer genes, and its meningeal tropism.

We did not observe salmochelin (*iro*) or aerobactin (*iuc*) gene clusters or hypermucoviscous *Klebsiella*-associated genes—*rmpADC*, *rmpA2*, and *magA*—in this *K. pneumoniae* collection.

**Genetic location of accessory virulence factors.** We investigated the genetic location of virulence factors using mlplasmids software (see Data Set S6 at https://doi.org/10.6084/m9.figshare.22794329.v2). The analysis predicted all core virulence gene clusters in chromosomal contigs (posterior probability ≥ 0.7). The accessory *kfuABC* gene cluster and the *wcaG* gene were found integrated into the chromosome (Fig. 4; see also Data Set S6). All PAI-derived yersiniabactin gene clusters were predicted as

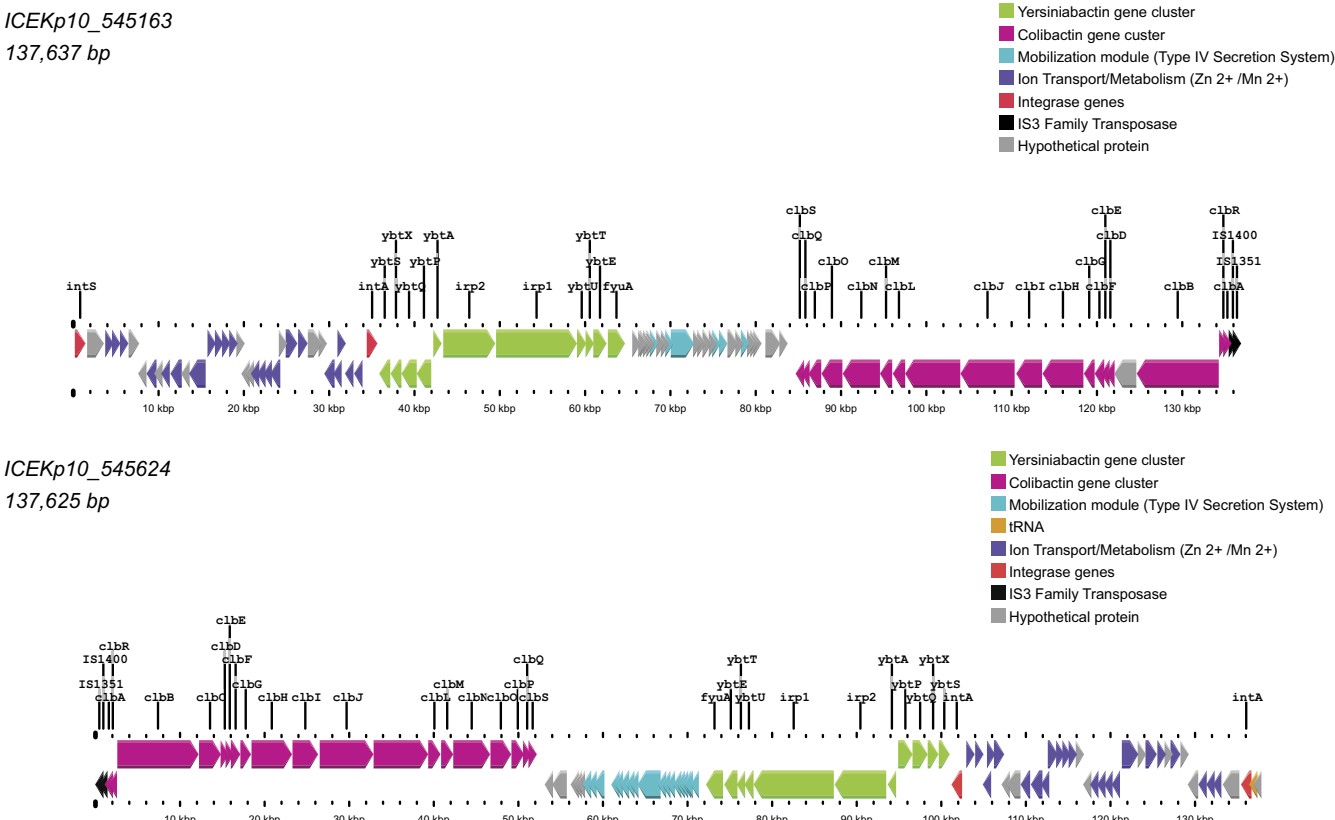

**FIG 5** ICE*Kp10* genomic structure reconstruction from hybrid assemblies, 545163 and 545624 isolates, using the Proksee web service. Two integrase genes (red) responsible for chromosomal integration and an additional $Zn^{+2}/Mn^{+2}$ ion uptake and metabolism module (purple) are shown. The mobilization module (blue), a type IV secretion system (T4SS), between the yersiniabactin (green) and colibactin (maroon) gene clusters are indicated. Two transposons (black) from IS*3* family transposase—IS*1400* originated from *Yersinia* bacteria and IS*351* from *Salmonella* based on the search using ISFinder (80)—were detected next to the colibactin gene cluster, which contribute to DNA transposition. Gray is used to indicate hypothetical proteins.

integrated into the chromosome. Colibactin gene clusters were also predicted in chromosomal contigs, as previously reported (53, 60).

**Other antimicrobial resistance and plasmid replicon genes in OXA-48- producing *K. pneumoniae* clinical isolates.** Five isolates had carbapenemases in addition to $bla_{OXA-48}$ gene. Four isolates from the Netherlands also carried a $bla_{NDM-1}$: two KL2 isolates (ST14), one KL17 isolate (ST101), and one KL51 isolate (ST16). One isolate from Spain (ST11, KL13) had also a $bla_{KPC-2}$ (see Fig. S6).

Besides carbapenemases, $bla_{CTX-M-15}$ was the most common ESBL gene in this study, a common phenomenon previously described (61), comprising 72% (82/114) of total isolates, 81% (43/53) among isolates from Spain and 64% (39/61) among isolates from the Netherlands. An initial analysis using mlplasmids predicted 61% (50/82) of the $bla_{CTX-M-15}$ genes in plasmid contigs with a posterior probability of >0.7, and 12% (10/82) were predicted in plasmid contigs but below the threshold (posterior probability, 0.5 to 0.6) (see Data Set S7 at https://doi.org/10.6084/m9.figshare.22794338.v2). However, 27% (22/82) of the remaining $bla_{CTX-M-15}$ genes were predicted in chromosomal contigs (posterior probability, <0.5) (see Table S4 at https://doi.org/10.6084/m9.figshare.22794398.v2). These 22 isolates with unexpected results for the location of $bla_{CTX-M-15}$ were additionally analyzed with RFplasmid software, and this tool predicted 19 of them in plasmid contigs (votes > 0.6 to 0.9), while three (3.6%, 3/82) were still not explicitly voted as plasmid-predicted contigs (see Table S4 and Fig. S6 at https://doi.org/10.6084/m9.figshare.22794530.v3).

We found other common antibiotic resistance genes: (i) the broad-spectrum $\beta$-lactamase $bla_{TEM-1B}$ (62/114, 54.4%); (ii) the oxacillin-hydrolyzing class D $\beta$-lactamase $bla_{OXA-1}$ (71/114, 62.3%); (iii) aminoglycoside phosphotransferase genes, mostly dominated by *aph*

$(3')$-*Ib*/*strA* (63/114, 55.3%) and *aph(6')*-*Id*/*strB* (60/114, 52.6%); (iv) aminoglycoside acetyl-transferase genes, the most prevalent being *aac(6)*-*Ib*-*cr* (68/114, 59.6%), followed by *aac(3)*-*IIa* (47/114, 41.2%); (v) quinolone resistance determinants such as *qnrB1* (38/114, 33.3%); (vi) sulfonamide-resistant genes, being *sul2* the most prevalent (60/114, 52.6%); (vii) the trimethoprim-resistant gene *dfrA14* (65/114, 57%); and (viii) the tetracycline resistance genes *tet(A)* (32/114, 28.1%) and *tet(D)* (23/114, 20.2%). All of these genes were predicted in plasmid contigs, except one $bla_{OXA-1}$ that was additionally analyzed using RFplasmid and then predicted in a plasmid contig (votes > 0.6) (see Fig. S6 and Data Set S7).

Among antibiotic resistance genes predicted in chromosomal contigs (see Fig. S7 at https://doi.org/10.6084/m9.figshare.22794593.v3 and Data Set S7 at https://doi.org/10.6084/m9.figshare.22794338.v2), most were from the $bla_{SHV}$ family as a common constituent of the *K. pneumoniae* chromosome (62), with $bla_{SHV-106}$ being the most common (36%, 41/114).

Regarding plasmid replicon genes, all OXA-48-*Kp* had the Inc/L plasmid replicon type. Four isolates additionally harboring a $bla_{NDM-1}$ gene had IncFIB and IncHI1B (ST14, KL2), IncA/C2 (ST16, KL51), and IncR (ST101, KL17) plasmid replicon types, which have been previously described associated with $bla_{NDM-1}$ (63–65). One isolate additionally harboring a $bla_{KPC-2}$ gene (ST11, KL13) had the IncP6 plasmid replicon type, as previously described (66) (see Data Set S8 at https://doi.org/10.6084/m9.figshare.22794359.v3 and Fig. S8 at https://doi.org/10.6084/m9.figshare.22794632.v3).

**Virulence and antibiotic resistance scores.** Following Lam et al. (17) criteria, the *K. pneumoniae* species complex can be scored based on the accumulation of clinically relevant antibiotic resistance and virulence loci. Virulence scores ranging from 0 to 5 are assigned depending on the presence of key virulence loci associated with increasing risk: yersiniabactin < colibactin < aerobactin. Resistance scores ranging from 0 to 3 are assigned based on antibiotic resistance genotypes and their escalation of antibiotic therapy: ESBL < carbapenemase < carbapenemase plus colistin resistance (17). Convergence of hypervirulence and antibiotic resistance is defined on the basis of these resistance and virulence scores: a virulence score of ≥3 (at least the *iuc* aerobactin gene cluster detected) and a resistance score of ≥1 (at least an ESBL gene detected). In our study, most isolates had a virulence score 1 (74/114, 65%) related to the presence of the yersiniabactin gene cluster, as expected for opportunistic *K. pneumoniae* infections (17), and a resistance score 2 (107/114, 94%) because of the carbapenemase presence. No convergence of hypervirulence and resistance was observed. Two isolates belonging to ST133 and ST792 had the highest virulence score of 2 due to the presence of colibactin and yersiniabactin gene clusters (see Data Set S5).

**Conclusions.** The majority of OXA-48-producing *K. pneumoniae* isolates had a pool of core chromosomally encoded virulence factors, such as the enterobactin gene cluster, fimbrial gene clusters, urea metabolism genes (*ureAD*), and the nonfimbrial adhesin *ycfM*. We did not observe carbapenem-resistant hypervirulent *K. pneumoniae* (CR-hv*KP*) in this collection, but two isolates had high virulence and a high resistance score due to the presence of colibactin gene cluster and the $bla_{OXA-48}$. Despite the high diversity of capsule locus among OXA-48-producing *K. pneumoniae*, some virulence gene clusters, such as *kfuABC* and *kpiABCDEFG* clusters, seemed to be associated with high-risk MDR clones: ST14, ST15, and ST405. Integrative conjugative elements (ICE*Kp*) were a predominant genetic structure for the spread of accessory virulence factors: the yersiniabactin gene cluster (ICE*Kp2*, ICE*Kp3*, ICE*Kp4*, ICE*Kp12*, and ICE*Kp22*) and the colibactin gene cluster (*ICEKp10*).

## MATERIALS AND METHODS

**Isolate collection.** WGS data of OXA-48-producing *K. pneumoniae* (OXA-48-*Kp*) clinical isolates from nationwide hospitals—sent to the Institute of Health Carlos III (ISCIII), Spain (ES) (*n* = 53, 44 participant hospitals), and the National Institute for Public Health and the Environment (RIVM), the Netherlands (NL) (*n* = 61, 31 participant hospitals)—were collected.

All OXA-48-*Kp*-NL isolates collected in 2016 and 2017 were included in this study, corresponding to 30 different sequence types (STs). We selected OXA-48-*Kp*-ES isolates based on predominant STs and common STs with NL collection, collected between 2011 and 2013. Common STs in both countries were

ST15 (ES, *n* = 12; NL, *n* = 3), ST101 (ES, *n* = 5; NL, *n* = 12), and ST147 (ES, *n* = 4; NL, *n* = 3). In addition, predominant STs were ST11 (ES, *n* = 14) and ST405 (ES, *n* = 16; NL, *n* = 1) from Spain and ST307 (NL, *n* = 8) from the Netherlands (see Fig. S1 at https://doi.org/10.6084/m9.figshare.22794056.v3). We excluded outbreak isolates to increase genomic diversity; these were isolates with a known epidemiological link, such as patient-to-patient transmission. Nevertheless, it is noteworthy the widespread presence of OXA-48-*Kp* in Spain (19).

Isolates were mostly from urine (ES, 64.2%, *n* = 34/53; NL, 23%, *n* = 14/61 [including two isolates from catheter]), blood (ES, 24.5%, *n* = 13/53; NL, 3.3%, *n* = 2/61), and perineum (NL, 19.6%, *n* = 12/61) or rectum (NL, 31.1%, *n* = 19/61). Less-common isolation sources included the respiratory system, wounds, and abscesses. One isolate from a Spanish patient and two isolates from Dutch patients had unknown sources (see Table S1 at https://doi.org/10.6084/m9.figshare.22794398.v2 and Fig. S1 at https://doi.org/10.6084/m9.figshare.22794056.v3).

**Whole-genome short-read sequencing and *de novo* assembly.** WGS was performed in each institution as follows: (i) ISCIII, DNA was extracted using the QIAamp DNA minikit (Qiagen, Hilden, Germany), genomic DNA paired-end libraries were generated using the Nextera XT DNA sample preparation kit (Illumina, Inc., San Diego, CA), and the libraries were sequenced using the Illumina NextSeq 500 sequencer system with 150-base paired-end reads (Illumina) and (ii) RIVM, DNA isolation and sequencing using the Illumina HiSeq 2500 was performed by using BaseClear B. V. (Leiden, the Netherlands), libraries were generated using a Nextera XT DNA sample preparation kit (Illumina), and sequencing yielded 150-base paired-end reads.

Short reads were quality trimmed and *de novo* assembled using Qiagen CLC Genomics Workbench software v7.0.4 (CLC Bio, Aarhus, Denmark), with the default parameters and a minimum contig length of 1,000 bp. Quality assessment of genome assemblies was done by using QUAST v5.0.2 (67). A quality assembly report is included as supplementary material (see Data Set S1 at https://doi.org/10.6084/m9.figshare.22794125.v3).

**Whole-genome long-read sequencing and *de novo* assembly.** Colibactin-positive OXA-48-*Kp* isolates were subjected to long-read sequencing using Oxford Nanopore Technologies (Oxford, UK). DNA extraction was done using an UltraClean microbial DNA isolation kit (Mo Bio Laboratories, Carlsbad, CA). Samples were barcoded with a native barcoding kit 1D (EXP-NBD103), and libraries were prepared without shearing to maximize sequencing read length using a ligation sequencing kit 1D (SQK-LSK108). Libraries were loaded onto a FLO-MIN106 R9.4 flow cell and run on a MinION device (48 h). Base calling was performed using Albacore v1.2.2. Long reads were *de novo* assembled using Unicycler v0.4.1. and combined with short reads (68) to obtain a hybrid assembly.

**Contamination screening.** All isolates were analyzed using Mash Screen v2.2.2, comparing a subset of 1,000 k-mers per sample against all NCBI RefSeq genomes (release 88) (69) to confirm the species identification, detect contamination with other species, and select a common reference for an efficient variant call (70).

**Core genome multilocus sequence typing.** A *Klebsiella pneumoniae/K. variicola/K. quasipneumoniae* cgMLST scheme consisting of 2,365 targets from SeqSphere+ (Ridom GmbH, Münster, Germany) was used for a gene-by-gene comparison of assembled genomes. This scheme was developed to ensure covering the genetic variability of the *K. pneumoniae* complex and a detailed description is available in the software. This cgMLST scheme was imported into chewBBACA 2.1.0 (71) to perform the analysis with the following parameters: a minimum BLAST score ratio for defining locus similarity at 0.6 and a minimum percentage of locus presence of 100% (default settings). The resulting matrix provided the loci that were present in 100% of the genomes (71) (see Data Set S2 at https://doi.org/10.6084/m9.figshare.22794197.v2). The geoBURST Full Minimum Spanning Tree was reconstructed using PHYLOViZ 2 (72).

According to the cgMLST SeqSphere+ server, closely related genomes are 'lumped' together in complex types (CTs; https://www.ridom.de/u/Core_Genome_MLST_Complex_Type.html) and a CT distance of 15 allele differences was applied as a first approximation to elucidate close genetic relatedness (https://www.cgmlst.org/ncs/schema/2187931/). This CT allele distance threshold is based on retrospective analysis of well-defined outbreaks and outgroup isolates with the same MLST/MLVA/PFGE profiles, as described elsewhere (73). In addition, we applied newly defined thresholds to capture the population structure of *K. pneumoniae*: a threshold of 43 allelic differences to identify *K. pneumoniae* clonal groups (CGs), and a threshold of 190 allelic differences to identify *K. pneumoniae* sublineages (SLs) (21). Genome assemblies were uploaded into Pathogenwatch global platform to obtain information on CGs and SLs (https://cgps.gitbook.io/pathogenwatch/) .

**Phylogenetic analysis using SNP distance.** We reconstructed a maximum-likelihood core genome SNP-phylogenetic tree, with general time reversible as the nucleotide substitution model, using MEGAX v10.1.7. (74). Considering only SNPs and ignoring other variant types and *K. pneumoniae* strain 19051 as the reference genome (accession number NZ_CP022023.1), core genome SNPs were obtained using Snippy v4.4.3 (https://github.com/tseemann/snippy), resulting in an alignment of 126,803 positions. The tree was visualized, together with available metadata, using Interactive Tree of Life iTOL v.6 (https://itol.embl.de/) (75).

**Capsular (K-Locus) and lipopolysaccharide (O-Locus) typing and other virulence factor screening.** First, we used Kleborate tool v2.3.0 for capsular typing (K-Locus) and lipopolysaccharide typing (O-Locus) and for identifying virulence and antibiotic resistance genotype scores, which can be used to infer hypervirulence and associated clinical risks (17).

Second, we built a local and customized database of 87 genes related to *Klebsiella* virulence based on a scientific literature review and *Klebsiella* curated public databases (see Table S2 at https://doi.org/10.6084/m9.figshare.22794398.v2). The database included (i) siderophores (enterobactin, *ent* and *fep* clusters; yersiniabactin, *ybt* gene cluster; salmochelin, *iro* gene cluster; aerobactin, *iuc* gene cluster); (ii)

fimbria synthesis (type 1 *fim* gene cluster and type 3 *mrk* gene cluster, and their associated gene *yfiN*); (iii) genes that are essential for cell physiology and survival such as urea (*ureAD*) and allantoin (*allS*) metabolism and also iron uptake system (*kfuABC*); (iv) genes for core LPS synthesis (*wabG* and *uge*), non-fimbrial adhesin (*ycfM*), and capsule modification (*wcaG*), as well as newly discovered fimbrial adhesin (*kpiABCDEFG*); and (v) mobile genetic derived genotoxin colibactin (*clb* gene cluster) and hypermucoviscosity-associated genes (*rmpA*, *rmpA2*, and *magA*). DNA sequences of these genes were collected from BIGSdb Institute Pasteur (https://bigsdb.pasteur.fr/klebsiella/) (48), NCBI GenBank, and the Virulence Factors Database (VFDB) (http://www.mgc.ac.cn/VFs/) (76, 77). OXA-48-*Kp de novo* assemblies—short-read assemblies except for colibactin positive isolates, 545163 and 545624, for which hybrid assemblies were considered—were screened using ABRicate 1.0.1 (https://github.com/tseemann/abricate) (80% gene coverage and 90% identity as thresholds). Further analysis of yersiniabactin and colibactin gene clusters was done using KLEBORATE v2.3.0 (https://github.com/katholt/Kleborate) (17). In addition, we analyzed ICEs harboring the colibactin gene cluster using the Proksee server (https://proksee.ca/) to create a linear representation.

**Antimicrobial resistance and plasmid replicon genes screening.** Short-read assembled genomes, as well as hybrid assemblies for 545163 and 545624 isolates, were screened for antibiotic resistance genes and plasmid replicon genes (80% coverage and 90% identity) using ABRicate 1.0.1 and ResFinder 3.1 (3,077 sequences, 13 May 2020 [last access]) and PlasmidFinder 2.1 (460 sequences, 13 May 2020 [last access]) databases (78, 79). ISFinder platform (https://isfinder.biotoul.fr) was considered for screening of insertion sequences (IS) in specific isolates (80).

**Genetic location of virulence factors, antibiotic resistance, and plasmid replicon genes.** The R package mlplasmids was used to predict each contig's chromosomal or plasmid class assignment. This tool is based on a Support Vector Machine model and provides posterior probabilities for contigs belonging to chromosome class or plasmid class. The classifier assigns each contig (assembly input) to the class with the highest posterior probability and considers a minimum contig length of 1,000 bp (81). A posterior probability threshold of 0.7 was further applied for chromosomal and plasmid class assignment certainty. mlplasmids and ABRicate results were combined using a Python script to predict the genetic location—chromosome or plasmid—of virulence, antimicrobial resistance, and plasmid replicon genes.

We used another tool for predicting plasmid sequences from short-read assembly data, RFplasmid (82), when unexpected results using mlplasmids were encountered, such as acquired antimicrobial resistance genes or plasmid replicon genes predicted in chromosomal contigs instead of plasmid contigs (plasmid posterior probability < 0.7 using mlplasmids).

**Data availability.** The raw reads analyzed for this study can be found under ENA BioProject PRJEB55414 (study ERP140306), run accession numbers ERR10775921 to ERR10776034 and experiment accession numbers ERX10227032 to ERX10227145.

## ACKNOWLEDGMENTS

We thank the Genomics Unit of the National Center of Microbiology (Madrid, Spain) for performing WGS for the isolates from Spanish patients.

A.P.J. and S.G.-C. conceived the study and drafted the manuscript. A.P.J., P.J.S.-C., M.P.-V., and S.G.-C. carried out the bioinformatics analysis. T.B., L.M.S., A.P.A.H., B.A., J.W.A.R., A.W.F., J.O.-I., B.A., and M.P.-V. helped interpreting the data, reviewed, and edited the manuscript. All authors revised and contributed to the article and approved the submitted version.

A.P.J. received an LPDP scholarship from the Ministry of Finance of the Republic of Indonesia for the financial support during the graduate study in University of Groningen. S.G.-C. received a fellowship (2018-T1/BMD-11581; MPY 337/19) awarded by the Dirección General de Investigación e Innovación, Consejería de Educación e Investigación, Comunidad de Madrid (CAM), and by the Instituto de Salud Carlos III (ISCIII), from the call "Ayudas destinadas a la Atracción de Talento Investigador 2018–Modalidad 1." This study was supported by Plan Nacional de I+D+i 2013-2016 and Instituto de Salud Carlos III, Subdirección General de Redes y Centros de Investigación Cooperativa, Ministerio de Economía, Industria y Competitividad, Spanish Network for Research in Infectious Diseases (REIPI RD16CIII/0004/0002), cofinanced by European Development Regional Fund ERDF "A Way To Achieve Europe," operative program Intelligent Growth 2014-2020. This study was also supported by a grant from the Instituto de Salud Carlos III (grant MPY 1135/16) and by the Antibiotic Resistance Surveillance Program of the Centro Nacional de Microbiología (Instituto de Salud Carlos III, Ministerio de Economía y Competitividad) of Spain. The Dutch CPE surveillance was funded by Dutch Ministry of Health, Welfare, and Sports.

The Dutch and Spanish Collaborative Working Groups on Surveillance on Carbapenemase-producing *Enterobacterales* (CPE) included (Netherlands) A. Maijer-Reuwer, ADRZ Medisch Centrum, Department of Medical Microbiology, Goes; M. A. Leversteijn-van Hall, Alrijne Hospital, Department of Medical Microbiology, Leiden; W. van den Bijllaardt, Amphia Hospital, Microvida Laboratory for Microbiology, Breda; R. van Mansfeld, Amsterdam UMC-Location

AMC, Department of Medical Microbiology, Amsterdam; K. van Dijk, Amsterdam UMC-Location Vumc, Department of Medical Microbiology and Infection Control, Amsterdam; B. Zwart, Atalmedial, Department of Medical Microbiology, Amsterdam; B. M. W. Diederen, Bravis Hospital/ZorgSaam Hospital Zeeuws-Vlaanderen, Department of Medical Microbiology, Roosendaal/Terneuzen; J. W. Dorigo-Zetsma, CBSL, Department of Medical Microbiology, Hilversum; D. W. Notermans, Centre for Infectious Disease Control, National Institute for Public Health and the Environment, Bilthoven; A. Ott, Certe, Department of Medical Microbiology, Groningen; W. Ang, Comicro, Department of Medical Microbiology, Hoorn; J. da Silvia, Deventer Hospital, Department of Medical Microbiology, Deventer; A. L. M. Vlek, Diakonessenhuis, Department of Medical Microbiology and Immunology, Utrecht; A. G. M. Buiting, Elisabeth-TweeSteden (ETZ) Hospital, Department of Medical Microbiology and Immunology, Tilburg; L. Bode, Erasmus University Medical Center, Department of Medical Microbiology, Rotterdam; S. Paltansing, Franciscus Gasthuis and Vlietland, Department of Medical Microbiology and Infection Control, Rotterdam; A. J. van Griethuysen, Gelderse Vallei Hospital, Department of Medical Microbiology, Ede; M. den Reijer, Gelre Hospitals, Department of Medical Microbiology and Infection prevention, Apeldoorn; M. J. C. A. van Trijp, Groene Hart Hospital, Department of Medical Microbiology and Infection Prevention, Gouda; M. Wong, Haga Hospital, Department of Medical Microbiology, 's-Gravenhage; A. E. Muller, HMC Westeinde Hospital, Department of Medical Microbiology, 's-Gravenhage; M. P. M. van der Linden, IJsselland hospital, Department of Medical Microbiology, Capelle a/d IJssel; M. van Rijn, Ikazia Hospital, Department of Medical Microbiology, Rotterdam; S. B. Debast, Isala Hospital, Laboratory of Medical Microbiology and Infectious Diseases, Zwolle; K. Waar, Certe Medische Microbiologie Friesland | Noordoostpolder, Department of Medical Microbiology, Leeuwarden; E. Kolwijck, Jeroen Bosch Hospital, Department of Medical Microbiology and Infection Control, 's-Hertogenbosch; N. Alnaiemi, LabMicTA, Regional Laboratory of Microbiology Twente Achterhoek, Hengelo; T. Schulin, Laurentius Hospital, Department of Medical Microbiology, Roermond; S. Dinant, Maasstad Hospital, Department of Medical Microbiology, Rotterdam; S. P. van Mens, Maastricht University Medical Centre, Department of Medical Microbiology, Maastricht; D. C. Melles, Meander Medical Center, Department of Medical Microbiology, Amersfoort; J. W. T. Cohen Stuart, Noordwest Ziekenhuisgroep, Department of Medical Microbiology, Alkmaar; M. L. van Ogtrop, OLVG Lab BV, Department of Medical Microbiology, Amsterdam; I. T. M. A. Overdevest, PAMM, Department of Medical Microbiology, Veldhoven; A. van Dam, Public Health Service, Public Health Laboratory, Amsterdam; I. Maat, Radboud University Medical Center, Department of Medical Microbiology, Nijmegen; B. Maraha, Albert Schweitzer Hospital, Department of Medical Microbiology, Dordrecht; J. C. Sinnige, Regional Laboratory of Public Health, Department of Medical Microbiology, Haarlem; E. E. Mattsson, Reinier de Graaf Groep, Department of Medical Microbiology, Delft; M. van Meer, Rijnstate Hospital, Laboratory for Medical Microbiology and Immunology, Velp; A. Stam, Saltro Diagnostic Centre, Department of Medical Microbiology, Utrecht; E. de Jong, Slingeland Hospital, Department of Medical Microbiology, Doetinchem; S.J. Vainio, St Antonius Hospital, Department of Medical Microbiology and Immunology, Nieuwegein; E. Heikens, St. Jansdal Hospital, Department of Medical Microbiology, Harderwijk; R. Steingrover, St. Maarten Laboratory Services, Department of Medical Microbiology, Cay Hill (St. Maarten); A. Troelstra, University Medical Center Utrecht, Department of Medical Microbiology, Utrecht; E. Bathoorn, University of Groningen, Department of Medical Microbiology, Groningen; T. A. M. Trienekens, VieCuri Medical Center, Department of Medical Microbiology, Venlo; D. W. van Dam, Zuyderland Medical Centre, Department of Medical Microbiology and Infection Control, Sittard-Geleen; E. I. G. B. de Brauwer, Zuyderland Medical Centre, Department of Medical Microbiology and Infection Control, Heerlen; Analytical Diagnostic Center N.V. Curaçao, Department of Medical Microbiology, Willemstad (Curacao); and Canisius Wilhelmina Hospital, Department of Medical Microbiology and Infectious Diseases, Nijmegen; and (Spain) Germán Bou, Complejo Hospitalario de A Coruña, A Coruña, and CIBERINFEC; Juan Manuel Hernández, Ana María Fernández Sánchez, Concepción Mediavilla Gradolph, Inmaculada de Toro Peinado, Hospital Regional Universitario de Málaga, Málaga; Ana María Fleites and Carlos Rodríguez-Lucas, Hospital Universitario Central de Asturias, Asturias; María Victoria García-López, Hospital

Clínico, Universidad de Málaga, Málaga; María Ángeles Orellana, Hospital 12 de Octubre, Madrid; Carmina Martí-Sala, Mª Angeles Pulido and Mayuli Armas, Hospital General de Granollers, Barcelona; Emilia Cercenado, Hospital General Universitario Gregorio Marañón, Madrid, and Centro de investigación en red de enfermedades respiratorias (CIBERES); Alejandro González-Praetorius and Sonia Solís, Hospital Universitario de Guadalajara, Guadalajara; María Isabel Sánchez-Romero, Hospital Universitario Puerta de Hierro, Majadahonda, Madrid; Rafael Cantón and Patricia Ruiz-Garbajosa, Hospital Universitario Ramón y Cajal, Madrid; Luis López-Urrutia Lorente, Hospital Universitario Río Hortega, Valladolid; Pedro de la Iglesia and Beatriz Iglesias, Hospital San Agustín de Avilés, Asturias; Guillermo Ruiz-Carrascoso, Hospital Universitario La Paz, Madrid; Mar Olga Pérez Moreno, Hospital de Tortosa Verge de la Cinta, Tortosa, Tarragona; Rosa Bartolomé and Juan José González-López, Hospital Universitario Vall d`Hebron, Barcelona; and CIBERINFEC; Carmen Conejo and Álvaro Pascual, Hospital Universitario Virgen Macarena, Sevilla, and CIBERINFEC; and José Antonio Rodríguez-Polo, Hospital Virgen de la Salud, Toledo.

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
