## [Reviewer comments · Microbiology Spectrum]

Microbiology Spectrum

Widespread detection of yersiniabactin gene cluster and its encoding integrative conjugative elements (ICE Kp) among non-outbreak OXA-48-producing *Klebsiella pneumoniae* clinical isolates from Spain and the Netherlands.

Afif Jati, Pedro J Sola-Campoy, Thijs Bosch, Leo Schouls, Antoni Hendrickx, Verónica Bautista, Noelia Lara, Erwin Raangs, Belen Aracil, John Rossen, Alex Friedrich, Ana Navarro, Javier Cañada-García, Eva Ramírez de Arellano, Jesús Oteo-Iglesias, María Pérez-Vázquez, and Silvia García-Cobos

Corresponding Author(s): Silvia García-Cobos, Instituto de Salud Carlos III Campus de Majadahonda

Review Timeline:

Submission Date:	November 18, 2022
Editorial Decision:	January 1, 2023
Revision Received:	May 19, 2023
Accepted:	May 22, 2023

Editor: Olaya Rendueles

Reviewer(s): Disclosure of reviewer identity is with reference to reviewer comments included in decision letter(s). The following individuals involved in review of your submission have agreed to reveal their identity: Andrés E Marcoleta (Reviewer #3)

Transaction Report:

DOI: <https://doi.org/10.1128/spectrum.04716-22>

January 1, 2023

Dr. Silvia García-Cobos
Instituto de Salud Carlos III Campus de Majadahonda
Ctra. Majadahonda-pozuelo Km.2
Majadahonda, Madrid 28220
Spain

Re: Spectrum04716-22 (**Convergence of virulence and carbapenem resistance in OXA-48-producing *Klebsiella pneumoniae* clinical isolates is mainly mediated by integrative conjugative elements spread**)

Dear Dr.García-Cobos, Dear Silvia:

Thank you for submitting your manuscript to Microbiology Spectrum.

Three reviewers have read through your manuscript and have highlighted several issues that should be addressed in depth and some ideas further developed in the discussion, before I can consider this manuscript for publication.

Specifically, please provide more detailed methods for the identification of the core genome.

Also, I agree with reviewer #2 and #3 that the general consensus for convergent isolates are those that have hypervirulence-associated loci (not only yersiniabactin) +carbapenemases.

Some conclusions are not fully supported by the data provided, please make sure that you provide all information required to prove your point.

To comply with the journal policies, make sure that all sequences are available in public repositories

Link Not Available

Sincerely,

Olaya Rendueles

Journals Department
Reviewer comments:

Reviewer #1 (Comments for the Author):

In this paper, Jati et al. investigated in silico the presence of virulence factors in OXA-48-positive isolates of *Klebsiella pneumoniae* isolated from Spain and The Netherlands. First, they studied the genetic relatedness and the distribution of K and O serotypes in the isolates. The authors described antigen epidemiology and showed how there is a higher diversity of K-antigens compared to O-antigens in *K. pneumoniae*. Then, they focused on describing the virulence factors in OXA-48-producing *K. pneumoniae*. These analyses included core and accessory genomes and finished with a description of antibiotic resistance genes in their strain collection.

The topic might be of interest to the Microbiology Spectrum audience. However, important issues need to be solved before publication. My main concern is that they do not provide enough evidence to make their claims. A good example is in the title which should be tuned down. Also, It is sometimes difficult to follow the text and during all the manuscript I could not follow the rationale in their analyses. Additionally, I would suggest to expand a bit the discussion as some issue such as discussing the highly association between OXA-48 and pOXA-48-like plasmids or the implications of studying the virulence in non-outbreak isolates.

Some sentences are hard to follow. i.e consider to shorten sentences like in 275-279.

Lines 268-269 Please provide statistical proof of their claim.

Lines 321-324 It is confusing how they include and exclude gene clusters in the core genome. It would be helpful for the reader to include a brief description regarding the definition of core virulence factors. Please include the size of the core-genome in Fig 2.

Lines 467-468 Please clarify in which conditions there is an evolutionary advantage. Is this common?

Line 477 pOXA-48 should be included now as incl. see Carattoli, A (2015). PLOS ONE
<https://doi.org/10.1371/journal.pone.0123063>

492-495 I suggest to remove this sentence as the role of *pemI/pemK* genes in chlorhexidine resistance is highly speculative.

Line 932 It would help the readers to include the criteria to choose the {less than or equal to}15 threshold.

Reviewer #2 (Comments for the Author):

In their paper, Jati et al provide an overview of the phylogenetic, antigenic and virulence properties of a collection of isolates from Netherlands and Spain, highlighting a separation of isolates from the two countries. Further, they also highlight widespread detection of yersiniabactin-encoding ICEKp elements in addition to other virulence loci (including those considered core to *K. pneumoniae*), flagging the 'convergence of virulence and resistance' in their isolates. Overall, the analyses presented in this study were sound, however the results don't add anything particularly novel to the existing literature. Widespread detection of the yersiniabactin-encoding ICEKp elements in *K. pneumoniae* isolates, including MDR clones, has already been previously documented. The term 'convergent virulent-resistant strains' (and similar) has generally been used to describe isolates that have both hypervirulence-associated loci (e.g. *iuc*, *iro*, *rmpADC/rmpA2*; those found on the virulence plasmid) and resistance to last line antimicrobials (e.g. carbapenems) and generally does not apply to isolates with yersiniabactin 'alone'. I also wanted to query the collection of isolates used in the study. The isolates from Netherlands vs Spain appeared to be collected from two different time periods, and therefore any conclusions/comparisons that are derived aren't particularly meaningful?

Other comments for consideration:

The authors used Gubbins to generate a recombination free alignment of core SNPs; however, as noted in the initial Gubbins publication by Croucher et al. 2014, recombination predictions are prone to false positives when the dataset encompasses non-clonal/divergent isolates from different lineages.

The authors state in lines 231 that the study encompasses non-outbreak isolates, however it is unclear from the methods/text what criteria the authors have used to define outbreak isolates.

Lines 239-241: the authors state here that they examine the phylogenetic relationship between strains in order to investigate 'virulence patterns', however given that most of the key virulence traits highlighted in the introduction are acquired via mobile elements, is it that useful to use core genome relatedness as the basis for examining virulence? Perhaps rephrase.

Line 241: how was this threshold derived?

Lines 260-279: by '34 combinations' are the authors referring to KL and OL combinations or KL+OL+ST combinations? I also wanted to flag the use of 'serotype' and reference to 'K' and 'O' types; Kaptive outputs the best matching K-locus and O-locus for each genome, which is generally predictive of serotype. It would therefore be more accurate to refer to the results as K/O loci instead of serotype and K types as 'KL'.

Line 316-317: It is interesting/unusual that *entD* was missing in almost 50% of their strains; did the authors verify this with read-mapping i.e. this can be done with SRST2.

Line 374: Note that *rmpA* is part of an operon, *rmpADC*

Lines 384-385: to clarify, ICEKp corresponds to the 'type' of integrative conjugative element, while ybt X (i.e. ybt 1) corresponds to the lineage.

Lines 406-407: what is the nucleotide divergence between these two CbSTs?

Line 408: Rephrase. The authors state that 'the K types are not described as hypervirulent...'; it is important to note here that disease pathotype/infection is a key defining attribute of a 'hypervirulent' strain, and that the presence alone of a particular capsule type/virulence loci does not define an isolate as being hypervirulent.

Minor comments:

Line 69: typo; missing 'of' between emergence and carbapenem

Line 71: unclear what is meant by 'mask' phenotype (for someone with a non clinical/microbiology background)

Line 120: for some context, how many hospitals does this involve?

Line 123: can the authors clarify the time period of isolate collection (i.e. unclear whether study period encompasses all of 2016 and 2017?)

Line 125: what are the six STs?

Lines 126-129: this text reads like it belongs in the results section rather than the methods

Line 205: what version of Kleborate?

Line 242: how many singletons were observed? How many of the isolates belonged to one of the 12 groups?

Line 248: which years?

Line 249: should this say eight groups?

Figure 1: would perhaps be useful to annotate on the figure the number of cgSNPs that differentiate isolates within each group. Additionally, different node symbols could be used to indicate the year in which the isolate was collected.

Reviewer #3 (Comments for the Author):

This article focuses on the genome sequencing and analysis of blaOXA-48-producing *K. pneumoniae*, a critical priority pathogen, isolated in Spain and The Netherlands. Overall, the article is well-written and easy to follow, and the methods are appropriate and sufficiently described. Also, the results are described in detail and support most of the conclusions. Nevertheless, in my opinion, there are some points the authors should address:

1) Please provide deposit/accession numbers for genome sequence data.

2) In the "Importance" section, it is stated that OXA-48 is the most prevalent carbapenemase in Europe, but no further explanations or citations were found. Please clarify.

3) Methods:

-line 152: what means "colibactin-producing positive OXA-48-Kp isolates"?

-line 156: verify that the code of the Ligation Sequencing Kit is correct (or is SQK-LSK109?)

4) cgMLST/cgSNP analysis and clone definition (methods and result). The relatedness of the isolates was evaluated by grouping the genomes according to the allele differences upon using a pre-defined cgMLST scheme for some *Klebsiella* species (cgMLST.org). A threshold of 15 allelic differences was established for grouping, corresponding to a 0.6% allelic distance and up to 10 cgSNPs. This is a very short distance, and more explanations should be provided regarding why it was chosen. Moreover, the grouping with such a strict threshold produced mostly singletons (Fig 1) and could hinder some relevant connections between isolates (possibly making the analysis more informative). I suggest examining the distribution of the pairwise allelic distances as an alternative to find appropriate thresholds suited to the dataset used. Bialek-Davenet et al (<https://www.ncbi.nlm.nih.gov/pmc/articles/PMC4214299/>) used this strategy to define *K. pneumoniae* clonal groups based on cgMLST and found a roughly 14% of allelic mismatches as the threshold to delineate each group.

5) Despite according to the results, none of the isolates harbored the typical markers of hypervirulent *K. pneumoniae* (aerobactin, rmp, salmochelin, virulence plasmid, allS, etc), in my opinion, some parts of the results and conclusions sections of the manuscript could be misleading, as the word convergence is explained in the context of hypervirulent Kp and then used to describe some of the isolates (which have no evidence of being hypervirulent). For instance, as stated in the text, K2 capsule not necessarily mean hypervirulence, but then in lines 507-509, it is referred to as a hypervirulent serotype. Moreover, since there are no experimental virulence data for these isolates, it is difficult to predict which one is more virulent than the others. In this context, it is unclear how kfuABC and KpiABCDEFG could be considered high-risk clone markers (lines 504-507). This statement should be explained/supported better.

Staff Comments:

Preparing Revision Guidelines

Please return the manuscript within 60 days; if you cannot complete the modification within this time period, please contact me. If you do not wish to modify the manuscript and prefer to submit it to another journal, please notify me of your decision immediately so that the manuscript may be formally withdrawn from consideration by Microbiology Spectrum.

Response to Reviewers

Spectrum04716-22: Convergence of virulence and carbapenem resistance in OXA-48-producing *Klebsiella pneumoniae* clinical isolates is mainly mediated by integrative conjugative elements spread.

Lines mentioned in our response are referred to the revised manuscript.

All modifications are highlighted in yellow in the Marked – up manuscript.

Authors' response to reviewers' comments

Reviewer #1:

In this paper, Jati *et al.* investigated *in silico* the presence of virulence factors in OXA-48-positive isolates of *Klebsiella pneumoniae* isolated from Spain and The Netherlands. First, they studied the genetic relatedness and the distribution of K and O serotypes in the isolates. The authors described antigen epidemiology and showed how there is a higher diversity of K-antigens compared to O-antigens in *K. pneumoniae*. Then, they focused on describing the virulence factors in OXA-48-producing *K. pneumoniae*. These analyses included core and accessory genomes and finished with a description of antibiotic resistance genes in their strain collection.

The topic might be of interest to the Microbiology Spectrum audience. However, important issues need to be solved before publication. My main concern is that they do not provide enough evidence to make their claims. A good example is in the title which should be tuned down. Also, it is sometimes difficult to follow the text and during all the manuscript I could not follow the rationale in their analyses. Additionally, I would suggest to expand a bit the discussion as some issue such as discussing the highly association between OXA-48 and pOXA-48-like plasmids or the implications of studying the virulence in non-outbreak isolates.

- **We thank Reviewer #1 for noticing us the lack of evidence to claim convergence events - AMR and hypervirulence - in our sample collection. We reviewed the “convergence” definition in this context and we have modified the title and the text accordingly. According to Lam et al. (Nature Communications, 2021), convergence is defined on the basis of resistance and virulence scores: virulence score ≥ 3 (at least the *iuc* aerobactin gene cluster detected) and resistance score ≥ 1 (at least an ESBL gene detected). We run the last version of the *Kleborate* tool and update these scores for our isolate collection.**

We changed our title to “Widespread detection of yersiniabactin gene cluster and its encoding integrative conjugative elements (ICEKp) among non-outbreak OXA-48-producing *Klebsiella pneumoniae* clinical isolates from Spain and the Netherlands.”

In addition, previous section 3.7 “Convergence of antibiotic resistance and virulence in OXA-48- producing *K. pneumoniae* clinical isolates” has been replaced and we now describe virulence and resistance scores used for hypervirulence definition (lines 464-477).

We have now emphasized the importance of studying non-outbreak isolates in the “Importance” section (lines 57-68).

We agree on the importance of OXA-48 carrying plasmids, nevertheless this has been described in other scientific papers and it is out of this study scope, in which we decided to focus on the description of virulence factors, and their concurrence with antibiotic resistance genes.

Some sentences are hard to follow. i.e consider to shorten sentences like in 275-279.

- **We have thoroughly reviewed the text and have modified the writing when needed; all changes are highlighted in yellow in the marked-up manuscript.**

Lines 268-269 Please provide statistical proof of their claim.

- **The sentence has been removed. The large variety of K/O loci combinations and sources makes it difficult to establish association, as it is observed in Figure 3.**

Lines 321-324 It is confusing how they include and exclude gene clusters in the core genome. It would be helpful for the reader to include a brief description regarding the definition of core virulence factors.

- **Section 3.3 has been modified for a better understanding. First, a brief description of previously well-defined core virulence factors in *K. pneumoniae* complex is given, and then possible additional core virulence factors are described based on the high presence (99-100%) of encoding genes in our studied collection and their function (lines 317-352).**

Please include the size of the core-genome in Fig 2.

- **The core-genome size (126,803bp) obtained from the new SNP analysis (without Gubbins as recommended by Reviewer #2 and following criteria by Croucher et al. 2014) is now indicated in the text (lines 188-189) and in Figure 2 legend.**

Lines 467-468 Please clarify in which conditions there is an evolutionary advantage. Is this common?

- **Following reviewer’s and editor’s recommendation about the lack of evidence for convergence events in our study, we have modified previous section 3.7 and highlighted instead the main findings regarding virulence and resistance scores (lines 464-477).**

Line 477 pOXA-48 should be included now as incl. see Carattoli, A (2015). PLOS ONE
<https://doi.org/10.1371/journal.pone.0123063>

- **Please see the response above. The corresponding sentence has been deleted.**

Lines 492-495 I suggest to remove this sentence as the role of pemI/pemK genes in chlorhexidine resistance is highly speculative.

- **We agree with the reviewer, the sentence has been removed.**

Line 932 It would help the readers to include the criteria to choose the {less than or equal to}15 threshold.

- **Further description about the cgMLST scheme and allele distance threshold, has been added in Material and Methods section (lines 173-182).**

Reviewer #2:

In their paper, Jati et al provide an overview of the phylogenetic, antigenic and virulence properties of a collection of isolates from Netherlands and Spain, highlighting a separation of isolates from the two countries. Further, they also highlight widespread detection of yersiniabactin-encoding ICEKp elements in addition to other virulence loci (including those considered core to *K. pneumoniae*), flagging the 'convergence of virulence and resistance' in their isolates. Overall, the analyses presented in this study were sound, however the results don't add anything particularly novel to the existing literature. Widespread detection of the yersiniabactin-encoding ICEKp elements in *K. pneumoniae* isolates, including MDR clones, has already been previously documented. The term 'convergent virulent-resistant strains' (and similar) has generally been used to describe isolates that have both hypervirulence-associated loci (e.g. *iuc*, *iro*, *rmpADC/rmpA2*; those found on the virulence plasmid) and resistance to last line antimicrobials (e.g. carbapenems) and generally does not apply to isolates with yersiniabactin 'alone'. I also wanted to query the collection of isolates used in the study. The isolates from Netherlands vs Spain appeared to be collected from two different time periods, and therefore any conclusions/comparisons that are derived aren't particularly meaningful?

- **We thank Reviewer #2 for his/her time reviewing our manuscript. We reviewed the term “convergent” and its definition in this context of hypervirulent-resistant *K. pneumoniae* isolates, and we have now changed the title and the text accordingly since none of the studied isolates fulfills such criterion.**

The novelty of this study resides in studying the virulent content of a widespread collection of OXA-48- producing *K. pneumoniae*. We aimed to capture the diversity of this bacteria involved in human infections by investigating non-outbreak related isolates.

We acknowledge the different periods of isolation, nevertheless we do not see any particular advantage in studying isolates from the same time period for our purpose. We focused on identifying possible virulence patterns in different STs of OXA-48- producing *K. pneumoniae* clinical isolates and we sought a broad collection to have a better overview of *K. pneumoniae* population structure.

Other comments for consideration:

The authors used Gubbins to generate a recombination free alignment of core SNPs; however, as noted in the initial Gubbins publication by Croucher et al. 2014, recombination predictions are prone to false positives when the dataset encompasses non-clonal/divergent isolates from different lineages.

- **We thank Reviewer’s comment, we have now used an alignment of core SNPs without recombination removal. The new number of SNPs are shown in table S3 and in figure 2, which represents the new phylogenetic tree without Gubbins analysis.**

The authors state in lines 231 that the study encompasses non-outbreak isolates, however it is unclear from the methods/text what criteria the authors have used to define outbreak isolates.

- **Isolates were selected based on clinical and epidemiological data, none of the included isolates had an epidemiological link indicating patient to patient transmission. This has been clarified in the text (lines 127-130).**

Lines 239-241: the authors state here that they examine the phylogenetic relationship between strains in order to investigate 'virulence patterns', however given that most of the key virulence traits highlighted in the introduction are acquired via mobile elements, is it that useful to use core genome relatedness as the basis for examining virulence? Perhaps rephrase.

- **We performed phylogenetic analysis to investigate possible genetic relatedness of the OXA-48-K. *pneumoniae* isolates included in the study and to ensure enough genetic diversity. We could expect to have a more similar accessory genome among very clonal isolates, due to horizontal transfer events, than among divergent isolates. Thus, we pursued a diverse collection of isolates to search for possible virulence patterns related to the ST/lineage itself, as we have observed with the *Kpi* operon. A better explanation is now given in section 3.1 (lines 241-265).**

Line 241: how was this threshold derived?

- **Further description about the cgMLST scheme and allele distance threshold has been added in Material and Methods section (lines 173-182). This threshold of 15 allele differences was established during the development of the cgMLST scheme used in RidomSeqSphere software, it was based on retrospective analysis of well-defined outbreaks and out-group isolates with the same MLST/MLVA/PFGE profiles (*Front Microbiol.* 2020 Jan 31;10:2961. doi: 10.3389/fmicb.2019.02961.). We applied it as a first approach to elucidate the presence of close genetically related isolates in our collection.**

Lines 260-279: by '34 combinations' are the authors referring to KL and OL combinations or KL+OL+ST combinations? I also wanted to flag the use of 'serotype' and reference to 'K' and 'O' types; Kaptive outputs the best matching K-locus and O-locus for each genome, which is generally predictive of serotype. It would therefore be more accurate to refer to the results as K/O loci instead of serotype and K types as 'KL'.

- **We referred to KL and OL combinations, we clarified it in the text (lines 270-271). We also modified the text using KL, O and K/O loci instead of serotypes.**

Line 316-317: It is interesting/unusual that *entD* was missing in almost 50% of their strains; did the authors verify this with read-mapping i.e. this can be done with SRST2.

- **We checked the percentage coverage for *entD* gene and we found that it was present but with a lower coverage (75% coverage) than the initial selected threshold (80%). This has been clarified in the revised manuscript (lines 325-326).**

Line 374: Note that *rpmA* is part of an operon, *rpmADC*

- **The text has been modified accordingly (line 378 and line 419).**

Lines 384-385: to clarify, ICEKp corresponds to the 'type' of integrative conjugative element, while *ybt X* (i.e. *ybt 1*) corresponds to the lineage.

- **We thank Reviewer's clarification. The text has been checked and corrected (lines 388-398).**

Lines 406-407: what is the nucleotide divergence between these two CbSTs?

- **There are three different alleles between clbST15 and clb19, please see below:**
ST: clbA, clbB, clbC, clbD, clbE, clbF, clbG, clbH, clbI, clbL, clbM, clbN, clbO, clbP, clbQ, clb

lineage.

clbST15: 2, 10, 2, 2, 2, 2, 2, 3, 2, 2, 2, 2, 6, 2, 2, clb3

clbST19: 2, 10, 2, 2, 2, 4, 2, 8, 2, 2, 2, 2, 2, 2, 2, clb3

This information has been extracted from: Search by locus combinations (pasteur.fr)

Line 408: Rephrase. The authors state that 'the K types are not described as hypervirulent...'; it is important to note here that disease pathotype/infection is a key defining attribute of a 'hypervirulent' strain, and that the presence alone of a particular capsule type/virulence loci does not define an isolate as being hypervirulent.

- **We agree with Reviewer's comment, the sentence has been removed.**

Minor comments:

Line 69: typo; missing 'of' between emergence and carbapenem

- **The error typo has been corrected (lines 72)**

Line 71: unclear what is meant by 'mask' phenotype (for someone with a non clinical/microbiology background)

- **OXA-48-producing strains can exhibit low-level carbapenem resistance, as can ESBL-producing strains with decreased permeability, which makes it difficult to detect OXA-48 production only based on the antibiotic susceptibility phenotype. A sentence has been added to clarify the term (lines 74-76).**

Line 120: for some context, how many hospitals does this involve?

- **The bacterial collection encompasses forty-four hospitals from Spain and thirty-one from The Netherlands. This information has been added in the manuscript (lines 118-121).**

Line 123: can the authors clarify the time period of isolate collection (i.e. unclear whether study period encompasses all of 2016 and 2017?)

- **Yes, both years. This has been clarified in the text (lines 122)**

Line 125: what are the six STs?

- **The paragraph has been modified for a better understanding (lines 122-130)**

Lines 126-129: this text reads like it belongs in the results section rather than the methods

- **We described the collection of isolates included in the study and we consider this more appropriate in the Material and Methods section than in the Results section.**

Line 205: what version of Kleborate?

- **Kleborate version v2.3.0, this is now indicated in the text (lines 193)**

Line 242: how many singletons were observed? How many of the isolates belonged to one of the 12 groups?

- **We observed seventy-four singletons, thus 40 isolates belonged to one of the 12 groups (complex types). This has been clarified in the text (lines 247-260).**

Line 248: which years?

- **Years 2016 and 2017. This has been clarified in the text (line 251).**

Line 249: should this say eight groups?

- **Section 3.1 has been modified and the referred sentence is now removed.**

Figure 1: would perhaps be useful to annotate on the figure the number of cgSNPs that differentiate isolates within each group. Additionally, different node symbols could be used to indicate the year in which the isolate was collected.

- **The number of cgSNPs is shown now in figure 2. Figure 2 has been modified using a core genome alignment without removing possible recombination (no Gubbins analysis). In addition, the number of allele differences and the number of cgSNPs are shown in supplementary table S3.**
- **Figure 1 displays now three different Minimum Spanning Trees based on three thresholds: ≤ 15 allele differences to identify Complex Types (CTs), ≤ 43 allele differences to identify Clonal Groups (CGs), and ≤ 190 allele differences to identify Sublineages (SLs). Further explanation is given in Material and Methods section (lines 173-182) and results (lines 243-265).**

Reviewer #3:

This article focuses on the genome sequencing and analysis of *bla*OXA-48-producing *K. pneumoniae*, a critical priority pathogen, isolated in Spain and The Netherlands. Overall, the article is well-written and easy to follow, and the methods are appropriate and sufficiently described. Also, the results are described in detail and support most of the conclusions. Nevertheless, in my opinion, there are some points the authors should address:

- **We thank Reviewer #3 for reviewing and considering our manuscript for publication.**

1) Please provide deposit/accession numbers for genome sequence data.

- **Project PRJEB55414 (Study ERP140306), accession numbers from ERR10775921 to ERR10776034. This information has been added in the manuscript (lines 838-839).**

2) In the "Importance" section, it is stated that OXA-48 is the most prevalent carbapenemase in Europe, but no further explanations or citations were found. Please clarify.

- **A new reference (reference 8) has been added (line 82): "Grudmann et al. Occurrence of carbapenemase-producing *Klebsiella pneumoniae* and *Escherichia coli* in the European survey of carbapenemase-producing Enterobacteriaceae (EuSCAPE): a prospective, multinational study. *The Lancet Infectious Diseases* 2017."**

3) Methods:

-line 152: what means "colibactin-producing positive OXA-48-Kp isolates"?

- **The sentence has been corrected as "colibactin-positive OXA-48-Kp isolates" (line 151).**

-line 156: verify that the code of the Ligation Sequencing Kit is correct (or is SQK-LSK109?)

- **The code is correct.**

4) cgMLST/cgSNP analysis and clone definition (methods and result). The relatedness of the isolates was evaluated by grouping the genomes according to the allele differences upon using a pre-defined cgMLST scheme for some *Klebsiella* species (cgMLST.org). A threshold of 15 allelic differences was established for grouping, corresponding to a 0.6% allelic distance and up to 10 cgSNPs. This is a very short distance, and more explanations should be provided regarding why it was chosen. Moreover, the grouping with such a strict threshold produced mostly singletons (Fig 1) and could hinder some relevant connections between isolates (possibly making the analysis more informative). I suggest examining the distribution of the pairwise allelic distances as an alternative to find appropriate thresholds suited to the dataset used. Bialek-Davenet et al (<https://www.ncbi.nlm.nih.gov/pmc/articles/PMC4214299/>) used this strategy to define *K. pneumoniae* clonal groups based on cgMLST and found a roughly 14% of allelic mismatches as the threshold to delineate each group.

- **The threshold of 15 allele differences was used to investigate close genetic relatedness of the isolates collection. We acknowledge it is a short distance but adequate for that purpose. Following Reviewer's recommendation and according to Bialek-Davenet et al, we looked at pairwise allelic distance. We observed a high number of genome pairs with <= 100 allele differences - corresponding to ~4% allelic distance in this study, cgMLST scheme of 2,365 targets - , just a few genome pairs having between 100-300 and the majority of genome pairs having far more than 300 allele differences. This observation suggests that roughly 100 targets/mismatches can be used to define clonal groups independently of the cgMLST size. In fact, considering a threshold <= 100 allele differences, we observed clonal groups mostly composed by isolates of the same ST.**

In this context, a recent publication (<https://doi.org/10.1093/molbev/msac135>) suggests the definition of new genetic dissimilarity thresholds: sublineages (SLs), threshold: 190 allelic mismatches; and clonal groups (CGs), threshold: 43 allelic mismatches. We additionally analyzed our collection using these new thresholds and described the obtained results in lines 261-265 and figures 1B and 1C.

5) Despite according to the results, none of the isolates harbored the typical markers of hypervirulent *K. pneumoniae* (aerobactin, rmp, salmochelin, virulence plasmid, allS, etc), in my opinion, some parts of the results and conclusions sections of the manuscript could be misleading, as the word convergence is explained in the context of hypervirulent Kp and then used to describe some of the isolates (which have no evidence of being hypervirulent). For instance, as stated in the text, K2 capsule not necessarily mean hypervirulence, but then in lines 507-509, it is referred to as a hypervirulent serotype. Moreover, since there are no experimental virulence data for these isolates, it is difficult to predict which one is more virulent than the others. In this context, it is unclear how kfuABC and KpiABCDEFG could be considered high-risk clone markers (lines 504-507). This statement should be explained/supported better.

- We reviewed the “convergence” definition in this context of hypervirulence and multidrug-resistance and we have modified the title and the text accordingly. Substantial modifications have been made in section 3.7, which is now focused on virulence and resistance scores obtained with Kleborate tool (lines 464-477).

Regarding *kfuABC* and *KpiABCDEFG* gene clusters, although we did not perform experimental virulence analysis in this study, this has been done elsewhere:

<https://doi.org/10.1073/pnas.1921393117> . Gato et al, demonstrated that *Kpi* contributes positively to the ability of *K. pneumoniae* to form biofilms and adhere to different host tissues. Nevertheless, the text has been revised and softened accordingly (lines 366-369). In addition, conclusions are now modified and the sentence “Carbapenem-resistant hypervirulent *K. pneumoniae* (CR-hvKP) due to hypervirulent serotypes acquiring plasmid-mediated resistance was anecdotic, with a K54 capsule strain and two K2 capsule strains harbouring a *blaOXA-48* gene” has been replaced (lines 481-488).

May 22, 2023

Dr. Silvia García-Cobos
Instituto de Salud Carlos III Campus de Majadahonda
Ctra. Majadahonda-pozuelo Km.2
Majadahonda, Madrid 28220
Spain

Re: Spectrum04716-22R1 (**Widespread detection of yersiniabactin gene cluster and its encoding integrative conjugative elements (ICEKp) among non-outbreak OXA-48-producing *Klebsiella pneumoniae* clinical isolates from Spain and the Netherlands.**)

Dear Dr. Silvia García-Cobos:

I am happy to inform you that your manuscript has been accepted, and I am forwarding it to the ASM Journals Department for publication. You will be notified when your proofs are ready to be viewed.

Sincerely,

Olaya Rendueles
Editor, Microbiology Spectrum
